# Interactive Cross-modal Learning for Text-3D Scene Retrieval

**Yanglin Feng[1], Yongxiang Li[1], Yuan Sun[2], Yang Qin[1], Dezhong Peng[1,3], Peng Hu[1]***

[1]College of Computer Science, Sichuan University, Chengdu, China.
[2]National Key Laboratory of Fundamental Algorithms and Models
for Engineering Numerical Simulation, Sichuan University, Chengdu, China.
[3]Tianfu Jincheng Laboratory, Chengdu, China.
`fcyzfyl@163.com, rhythmli.scu@gmail.com, sunyuan_work@163.com,`
`qinyang.gm@gmail.com, pengdz@scu.edu.cn, penghu.ml@gmail.com`

## Abstract

Text-3D Scene Retrieval (T3SR) aims to retrieve relevant scenes using linguistic queries. Although traditional T3SR methods have made significant progress in capturing fine-grained associations, they implicitly assume that query descriptions are information-complete. In practical deployments, however, limited by the capabilities of users and models, it is difficult or even impossible to directly obtain a perfect textual query suiting the entire scene and model, thereby leading to performance degradation. To address this issue, we propose a novel **I**nteractive Text-3**D** Scene Retrieval Method (IDeal), which promotes the enhancement of the alignment between texts and 3D scenes through continuous interaction. To achieve this, we present an Interactive Retrieval Refinement framework (IRR), which employs a questioner to pose contextually relevant questions to an answerer in successive rounds that either promote detailed probing or encourage exploratory divergence within scenes. Upon the iterative responses received from the answerer, IRR adopts a retriever to perform both feature-level and semantic-level information fusion, facilitating scene-level interaction and understanding for more precise re-rankings. To bridge the domain gap between queries and interactive texts, we propose an Interaction Adaptation Tuning strategy (IAT). IAT mitigates the discriminability and diversity risks among augmented text features that approximate the interaction text domain, achieving contrastive domain adaptation for our retriever. Extensive experimental results on three datasets demonstrate the superiority of IDeal. Code is available at `https://github.com/Yangl1nFeng/IDeal`.

## 1 Introduction

Recent years have witnessed natural language interfaces to embodied intelligence systems [1, 2, 3, 4, 5] become increasingly prevalent in our daily lives. This opens up further opportunities for natural language-based interaction with intelligent agents, such as a user verbally instructing agents to perform tasks in a specific scene. Before executing any task, an agent must first retrieve the scene relevant to the user's intent. This requirement has spurred recent works on Text-3D Scene Retrieval (T3SR) [6, 7], which enables the retrieval of 3D point-cloud scenes using linguistic queries. Such a language-scene alignment capability lays a critical foundation for enabling agents to generalize across scenes and environments.

Although existing dedicated methods [6] achieve promising performance for T3SR by facilitating fine-grained query text and scene understanding, such success often relies on the assumption that

---

*Corresponding author.

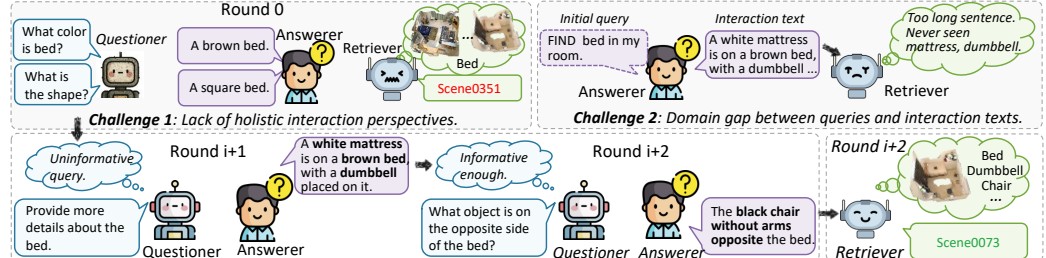

Figure 1: Overview of interactive text-3D scene retrieval. The above gray part illustrates the two specific challenges, while the below white part shows the illustrations of our interactive framework.

the queries provided are information-complete. However, such an assumption is often violated in real-world scenarios due to the inherent limitations of text inputs and models, such as incomplete one-shot descriptions of user intent [8], ambiguous descriptions [6], domain shifts [9], and limited generalization of the models. As a result, the performance and robustness of the models remain persistently constrained, and relying solely on limited internal knowledge is insufficient to overcome this inherent bottleneck.

To break through the bottleneck, recent studies [8, 10, 11] have explored integrating external knowledge from Large Language Models (LLMs) to enhance their understanding and alignment abilities. However, such approaches typically require prohibitively expensive fine-tuning or retraining of offline models [10, 12]. Some other attempts [8, 13, 14] have proposed interactive cross-modal retrieval frameworks that incorporate LLMs and Vision-Language Models (VLMs) to facilitate more fine-grained understanding and alignment, thereby iteratively evolving the retrieval performance. Although these methods have demonstrated remarkable effectiveness in image-text matching [8] and video-text retrieval [15, 16], they still face two tricky challenges in the T3SR setting, as shown in Figure 1. *Firstly*, since the scale and complexity of 3D scenes, these methods lack holistic perspectives beyond a localized focus during interaction, *e.g.*, the LLMs tend to focus on the salient objects in the scene and ignore the fine-grained details at a scene-level perspective, limiting the depth and breadth of LLM interaction, as demonstrated in Table 1. *Secondly*, existing retrieval models exhibit limited generalization ability to biased text domains, limiting their effectiveness in handling realistic interaction texts that exhibit domain gaps.

To address the aforementioned challenges, this paper proposes a novel **I**nteractive Text-3**D** Sc**e**ne Retriev**al** method (IDeal) to conduct continuous interaction between the T3SR models and external users (*e.g.*, LLMs), achieving the active alignment between text queries and 3D scenes, as depicted in Figure 1. Our IDeal consists of two components: an Interactive Retrieval Refinement framework (IRR) and an Interaction Adaptation Tuning strategy (IAT), as illustrated in Figure 2. More specifically, IRR coordinates three specialized agents (*i.e.*, *questioner*, *answerer*, and *retriever*) to perform multi-round interaction. First, the *questioner* adaptively determines whether to continue probing object details or to pursue divergence by exploring the broader scene, based on the assessment of the current round's description. Based on this, it continuously formulates context-relevant questions to the *answerer*. After receiving responses, the *retriever* iteratively integrates information at both the feature and semantic levels, facilitating comprehensive scene-level understanding for progressively precise re-rankings. To mitigate the domain shift between training queries and interactive texts, IAT proposes adapting the *retriever* toward the interaction text domain. Specifically, IAT leverages LLMs to generate more realistic augmented texts that closely resemble the interaction text domain. Subsequently, IAT robustly mitigates the discriminability and diversity theoretical risks in the features of the augmented texts for domain gap bridging, thereby ensuring an unbiased understanding of the interaction texts by the *retriever*. The contributions of this paper are as follows:

- We propose a novel Interactive Text-3D Scene Retrieval Method (IDeal), which actively enhances alignment between text queries and 3D scenes through ongoing interactions.
- An Interactive Retrieval Refinement framework (IRR) is presented to enable a deep interaction for comprehensive scene exploration, leading to progressively improved retrieval.
- An Interaction Adaptation Tuning strategy (IAT) is proposed, which facilitates the transfer of the retriever to the interaction text domain, promoting improved interaction.
- We conduct extensive comparison experiments on text-3D scene datasets. Our IDeal remarkably outperforms the existing methods, demonstrating its superiority.

## 2 Related Work

**Cross-Modal Retrieval.** Cross-Modal Retrieval (CMR) [17, 18, 19, 20, 21, 22] aims to match corresponding results across modalities for a given query, bridging the gap caused by modal heterogeneity. In recent years, CMR has garnered significant attention in fields such as Image-Text Retrieval [14, 23, 24, 25], Video-Text Retrieval [26, 27], 2D-3D Retrieval [28, 29], Pointcloud-Text Matching [6]. The primary challenge of CMR lies in effectively aligning multimodal data. To address this issue, most existing works could be broadly categorized into two groups: 1) Coarse-grained retrieval [30, 31, 32] directly maps multimodal data into a shared space, aiming for a more straightforward and computationally efficient alignment. 2) Fine-grained retrieval [33, 34] seeks to establish local associations between fine-grained features across modalities (*e.g.*, regions in images, words in texts). These local associations are then progressively integrated to form precise cross-modal correspondences. This paper focuses on a more challenging CMR task, *i.e.*, Text-3D Scene Retrieval, involving obscure spatial cues and sophisticated 3D scenes (including issues such as viewpoints and occlusions [35, 36]). Although prior work [6] performs well with comprehensive descriptions, it struggles with online and blurry queries. To this end, we propose an Interactive T3SR solution that iteratively incorporates online feedback to achieve more precise and practical scene retrieval.

**Interactive Learning.** Unlike traditional learning paradigms [6, 37], interactive learning emphasizes the continuous improvement of a model's behavior through ongoing interactions with the environment or users. Specifically, several pioneering works [38, 39] leverage simple forms of user feedback (*e.g.*, preferred sample selection and relevance scoring) to iteratively achieve improved training quality or better satisfy user-specific requirements during testing. With the development of Large Language Models (LLMs), other studies [8, 40, 41] have begun exploring question-answering interactions through free-form text dialogue, closely replicating natural human communication. For example, several methods leverage iterative interactions to continuously refine the retrieval query for better reranking. Recently, more advanced methods such as PlugIR [13], MERLIN [16], ICL [42], and LLaVA-ReID [43] have integrated LLMs for context-aware question generation, mining more visual details. However, these methods cannot be effectively generalized to T3SR due to the differences in tasks and data domains shown in Figure 1. In this paper, we develop an interactive framework tailored for T3SR to help the offline models adapt to complex scene perception.

## 3 Method

### 3.1 Problem Formulation

Given a text query set $\mathcal{T} = \{t_i\}_{i=1}^{n_t}$ and a 3D scene gallery $\mathcal{C} = \{c_j\}_{j=1}^{n_c}$, where $t_i$ and $c_j$ represent $i$-th text and $j$-th scene, $n_t = |\mathcal{T}|$ and $n_c = |\mathcal{C}|$ means the sample number, and $|\cdot|$ denotes the volume of set. The purpose of T3SR is to use the text query to match the ideal 3D scenes from the gallery, where there exists correspondence $y_{i,j} \in \{0, 1\}$, indicating whether the points are matched (*i.e.*, $y_{ij} = 1$), or unmatched (*i.e.*, $y_{ij} = 0$). Existing methods [6, 37] typically train an offline model to achieve encoding of multimodal data, followed by meticulous single-turn retrieval. They assume that user-provided text queries are information complete, overlooking the practical fact that queries are often partial, ambiguous, or even exhibit domain shift.

To address these issues, an interactive Text-3D scene retrieval method, *i.e.*, IDeal, is proposed to bridge the interaction between the retrieval models and external agents, progressively overcoming the aforementioned query limitations and achieving improved alignment between texts and 3D scenes. More specifically, IDeal asks question $q_i^l$ ($l \in \{1, \cdots, r\}$) about $i$-th sample based on the users' previous response $a_i^{l-1}$ ($a_i^0 = t_i$), where $r$ is the upper limit of rounds. Subsequently, the external agents recall details and answer $a_i^l$ of the target 3D scene $c_j$ ($y_{ij} = 1$), forming a dialogue context $\mathcal{D}_i = \{a_i^0, (q_i^1, a_i^1), \cdots (q_i^r, a_i^r)\}$ composed of question-answer pairs $(q_i, a_i)$. Both the $j$-round response text and 3D scenes are projected into a shared feature space by a trained retrieval model, which can be: $\boldsymbol{u}_i^j = f_r(a_i^j; \boldsymbol{\theta})$ and $\boldsymbol{v}_i = f_r(c_i; \boldsymbol{\theta})$, where $\boldsymbol{\theta}$ denotes the learnable parameters.

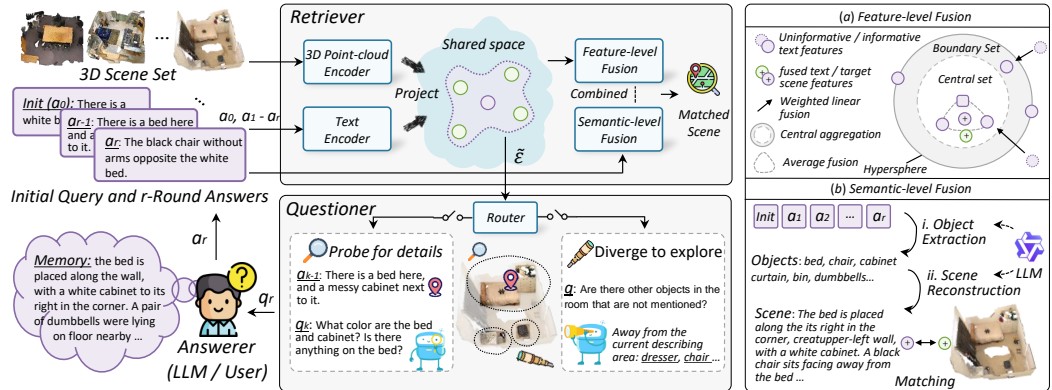

Figure 2: (Left) Pipeline of the proposed IDeal. A *questioner* employs a router to adaptively pose either probe or divergent questions, which require responses from an *answerer*. After receiving iterative responses, a *retriever* projects the multimodal data into a shared feature space, performing feature-/semantic-level fusion (Right) to enable progressively precise scene retrieval.

## 3.2 Interactive Retrieval Refinement Framework

In this section, we introduce the Interactive Retrieval Refinement framework (IRR), which coordinates three agents (*i.e.*, *questioner*, *answerer*, and *retriever*) to enable iterative interaction. In the following sections, we will elaborate on them by introducing their interaction process.

### 3.2.1 Adaptive Questioning

To achieve a deep interaction for T3SR, we present an adaptive *questioner*, which enables pertinent questioning to the *answerer* for both detailed and comprehensive exploration of complex 3D scenes. Firstly, a question router is employed, which determines the focus of questioning based on the feature distribution in the shared space. More specifically, the router assesses whether the previous-round description is informative by computing a *Cross-modal Affinity Entropy* ($\mathcal{E}$), formulated as:

$$\mathcal{E}(\boldsymbol{u}_i^{r-1}) = - \sum_{j \in N_k(\boldsymbol{u}_i^{r-1})} p(\boldsymbol{u}_i^{r-1}, \boldsymbol{v}_j) \log p(\boldsymbol{u}_i^{r-1}, \boldsymbol{v}_j), \tag{1}$$

where $N_k(\boldsymbol{u}_i^{r-1})$ means the $k$-nearest-neighbor index of $\boldsymbol{u}_i^{r-1}$, and affinity probability $p(\boldsymbol{u}_i^{r-1}, \boldsymbol{v}_j)$ is formulated as:

$$p(\boldsymbol{u}_i^{r-1}, \boldsymbol{v}_j) = \frac{\exp(\mathcal{S}(\boldsymbol{u}_i^{r-1}, \boldsymbol{v}_j)/\tau)}{\sum_{l \in N_k(\boldsymbol{u}_i^{r-1})} \exp(\mathcal{S}(\boldsymbol{u}_i^{r-1}, \boldsymbol{v}_l)/\tau)}, \tag{2}$$

where $\mathcal{S}$ represents the computation of similarity between features, and $\tau$ is a temperature parameter.

However, the distribution of scene features extracted by the trained retrieval models is fixed and inherently non-uniform, with regions of over-density and under-density introducing a structural density bias [44]. To mitigate the bias, we introduce a *Density Compensated Factor* for each scene feature, which is formulated as: $\rho(\boldsymbol{v}_i) = \frac{1}{(1/k) \sum_{j \in N_k(\boldsymbol{v}_j)} \mathcal{D}(\boldsymbol{v}_i, \boldsymbol{v}_j) + \epsilon}$, where $\mathcal{D}$ represents the distance calculation and $\epsilon$ is a minimal constant for numerical stability. Based on this, we try to approximately correct the original similarity score $\mathcal{S}(\boldsymbol{u}_i^{r-1}, \boldsymbol{v}_j)$ by incorporating the *Density Compensated Factor* $\rho(\boldsymbol{v}_i)$, which could be written as: $\tilde{\mathcal{S}}(\boldsymbol{u}_i^{(r-1)}, \boldsymbol{v}_j) = \mathcal{S}(\boldsymbol{u}_i^{(r-1)}, \boldsymbol{v}_j)/\sqrt{\rho(\boldsymbol{v}_j)}$. Subsequently, this corrected similarity is brought back into Equations (1) and (2) to obtain a *Density Compensated Affinity Entropy*, denoted as $\tilde{\mathcal{E}}$. This process approximates a Bayesian Correction leveraging prior density estimation[2], mitigating the impact of the inherent bias in scene feature distribution.

Leveraging the fairer metric $\tilde{\mathcal{E}}$, our *questioner* categorize descriptions with $\tilde{\mathcal{E}} > \beta$ as uninformative, prompting an LLM to generate *questions for detail probe* (*i.e.*, $\mathcal{Q}_1$) for attribute and spatial relationship detail refinements within the described area. Conversely, when $\tilde{\mathcal{E}} \leq \beta$, the descriptions are considered informative, triggering the adopting of *questions for diverge exploration* (*i.e.*, $\mathcal{Q}_2$) that inquire about object arrangements not previously discussed in the dialogue.

---

[2] Please refer to our Supplemental Material for further discussion.

### 3.2.2 Iterative Retrieval

After completing the questioning, an LLM is employed to simulate the external user acting as an *answerer* to answer the questions, following existing interactive approaches [8, 13]. It receives multi-round questions and provides responses based on its memory. In this paper, we adopt text modality to simulate the memory, which better approximates how humans recall information in mind.

Upon receiving the response descriptions from the *answerer*, we construct a *retriever* that can be seamlessly integrated with existing cross-modal models [37, 45], enabling iterative scene retrieval. It can obtain the scene retrieval predictions for any given text within the shared feature space. Specifically, given a text feature $\boldsymbol{u}_i$, the prediction can be formulated as:

$$\hat{\boldsymbol{p}}(\boldsymbol{u}_i) = [\hat{p}(\boldsymbol{u}_i, \boldsymbol{v}_1), \hat{p}(\boldsymbol{u}_i, \boldsymbol{v}_2), \ldots, \hat{p}(\boldsymbol{u}_i, \boldsymbol{v}_{n_c})]^\top, \tag{3}$$

where $\hat{p}(\boldsymbol{u}_i, \boldsymbol{v}_j) = \exp(\mathcal{S}(\boldsymbol{u}_i, \boldsymbol{v}_j))/\sum_{l=1}^{n_c} \exp(\mathcal{S}(\boldsymbol{u}_i, \boldsymbol{v}_l))$ denotes the probability that the $i$-th text retrieves the $j$-th scene. Accordingly, our *retriever* first utilizes the initial query to compute an *initial retrieval prediction* $\hat{\boldsymbol{p}}_1(\boldsymbol{u}_i)$. Subsequently, the feature-level and semantic-level fusion of interactive responses is conducted to achieve more precise scene retrieval.

For the fusion of interactive response feature, on one hand, considering responses to $\mathcal{Q}_1$ are refinements of the previous-round descriptions, we apply a weighted linear fusion to incorporate supplementary information. This strategy enables the preservation of core semantic cues from previous rounds while emphasizing newly introduced details: $\boldsymbol{u}_i^j = \alpha \boldsymbol{u}_i^j + (1-\alpha)\boldsymbol{u}_i^{j-1}$, where $q_i^j \in \mathcal{Q}_1$, $\alpha$ is a trade-off weight. On the other hand, benefiting from $\mathcal{Q}_2$, the other response features and the aforementioned fused features capture variations across different regions of the scene. To fuse them into a comprehensive feature, inspired by [46], we model their distribution around the target scene by encapsulating them within a minimum enclosing hypersphere:

$$(\boldsymbol{o}_i^*, R_i^*) = \arg\min_{\boldsymbol{o}_i, R_i} \left\{ R_i : \boldsymbol{u}_i^j \boldsymbol{o}_i^\top \leq R_i, \forall j \right\}, \tag{4}$$

where $\boldsymbol{o}_i^*$ and $R_i^*$ are the center and radius of the hypersphere, respectively. Based on this, features near the hypersphere boundary are grouped into a boundary set $\mathcal{U}_i^1$, while the remainder constitute the central set $\mathcal{U}_i^2$. To balance fusion robustness and feature discrimination, potentially noisy boundary features in $\mathcal{U}_i^1$ are aggregated at the hypersphere center and averaged with cleaner features in $\mathcal{U}_i^2$ to yield the final fused response feature: $\bar{\boldsymbol{u}}_i = \frac{1}{2}\left(\boldsymbol{o}_i^* + \frac{1}{|\mathcal{U}_i^2|}\sum_{\boldsymbol{u}_i^j \in \mathcal{U}_i^2} \boldsymbol{u}_i^j\right)$. This fused feature is then input into Equation (3) to obtain an *interactive feature prediction* $\hat{\boldsymbol{p}}_2(\bar{\boldsymbol{u}}_i)$.

However, the aforementioned feature-level fusion can not fully capture the holistic semantics of the responses. To address this limitation, we leverage an LLM to reconstruct the 3D scene from all responses in textual space. More specifically, inspired by Chain-of-Thought (CoT) [47], we decompose this process into object extraction and scene reconstruction for a more stable and comprehensive scene summary. The LLM first identifies the scene objects across multi-round responses and then summarizes an object-centric scene reconstruction text. Finally, the texts are encoded into feature $\boldsymbol{s}_i$, from which a *interactive semantic prediction* $\hat{\boldsymbol{p}}_3(\boldsymbol{s}_i)$ is computed using Equation (3).

Finally, the initial and interactive predictions are combined through weighted fusion to obtain the final scene retrieval prediction as follows:

$$\hat{\boldsymbol{p}}_c(\boldsymbol{u}_i) = \lambda_1 \hat{\boldsymbol{p}}_1(\boldsymbol{u}_i) + \lambda_2 \hat{\boldsymbol{p}}_2(\bar{\boldsymbol{u}}_i) + \lambda_3 \hat{\boldsymbol{p}}_3(\boldsymbol{s}_i), \tag{5}$$

where $\hat{\boldsymbol{p}}_c(\boldsymbol{u}_i)$ is the final retrieval prediction, $\lambda_1$, $\lambda_2$, and $\lambda_3$ are trade-off parameters. Benefiting from the adaptive questioning and the comprehensive retrieval information fusion, our IDeal can alleviate the limitations of initial queries through progressive interaction.

### 3.3 Interaction Adaptation Tuning

Although IRR can exploit interactions to promote retrieval quality, the limited text domain of the *retriever* remains a bottleneck that restricts further performance improvements. To overcome this limitation, we propose an Interaction Adaptation Tuning strategy (IAT), which enhances texts to approximate the domain of interaction texts.

We begin by integrating information and descriptions of the same scenes from the training data to construct simulated memory for text augmentation. Following IRR, we first provide a training-data-based answerer (*i.e.*, an LLM) with the constructed memory and initial queries. We then simulate the

IRR interaction process by iteratively posing a fixed number of $\mathcal{Q}_1$ and $\mathcal{Q}_2$ questions. The response descriptions yield augmented texts that closely approximate the interaction scenario.

After obtaining the enriched augmented texts, inspired by the contrastive domain adaptation paradigm [48, 49], we try to minimize the theoretical risk $\mathcal{R}(\boldsymbol{\theta})$ associated with our *retriever* among the augmented text features, as formulated below. It facilitates model adaptation of the *retriever* to the augmented text domain without requiring access to its implementation details.

$$\mathcal{R}(\boldsymbol{\theta}) = \mathcal{R}_{dis}(\boldsymbol{\theta}) + \mathcal{R}_{div}(\boldsymbol{\theta}) = \mathbb{E}_{\tilde{\mathcal{U}}} \left[ \left( -\mathbb{E}_{\tilde{\mathcal{U}}^+} \left\{ \mathcal{S}(\tilde{\boldsymbol{u}}_i^+, \tilde{\boldsymbol{u}}_i) \right\} \right) + \left( \mathbb{E}_{\tilde{\mathcal{U}}^-} \left\{ \mathcal{S}(\tilde{\boldsymbol{u}}_i^-, \tilde{\boldsymbol{u}}_i) \right\} \right) \right], \quad (6)$$

where the two components $\mathcal{R}_{dis}(\boldsymbol{\theta})$ and $\mathcal{R}_{div}(\boldsymbol{\theta})$ respectively reflect discriminability and diversity risks, $\mathbb{E}$ denotes expectation, $\mathbb{E}_{\tilde{\mathcal{U}}}$ is taken with respect to the distribution for the target domain features (*i.e.*, augmented text features $\tilde{\mathcal{U}} = \{\tilde{\boldsymbol{u}}_i\}_{i=1}^{n_t}$), and $\tilde{\mathcal{U}}^+$ and $\tilde{\mathcal{U}}^-$ is the distribution for the corresponding positive features $\tilde{\boldsymbol{u}}_i^+$ and negative features $\tilde{\boldsymbol{u}}_i^-$, respectively.

*On the one hand*, minimizing the discriminability risk $\mathcal{R}_{dis}(\boldsymbol{\theta})$ requires encouraging the augmented text features to align closely with those belonging to the same scenes. However, the scale and complexity of scenes often lead to substantial variability even among features corresponding to the same scenes. This introduces significant uncertainty in the selection of positive samples, complicating the risk optimization process.[3] To handle this, we adopt the corresponding 3D scene features as substitutes for the text features to construct positive pairs. This is based on the assumption that the scene features encoded by the well-trained model are more stably located near the center of the corresponding description distribution. Finally, we mitigate the aforementioned discriminability risk $\mathcal{R}_{dis}(\boldsymbol{\theta})$ by minimizing a negative log-based proxy loss term $\mathcal{L}_{dis}$, which could be written as follows:

$$\mathcal{L}_{dis} = -\sum_{i=1}^{b} \sum_{j=1}^{n_c} y_{ij} \log \mathcal{S}(\tilde{\boldsymbol{u}}_i, \boldsymbol{v}_j), \quad (7)$$

where $b$ is the size of the mini-batch. *On the other hand*, motivated by [50], we attempt to approximate the minimization of the divergence risk $\mathcal{R}_{div}(\boldsymbol{\theta})$ by minimizing its upper bound, *i.e.*,

$$\sup\left(\mathcal{R}_{div}(\boldsymbol{\theta})\right) \sim \left\{ \mathbb{E}_{\tilde{\boldsymbol{u}}^- \sim \tilde{\mathcal{U}}^-} \left( \mathcal{S}(\tilde{\boldsymbol{u}}, \tilde{\boldsymbol{u}}^-) \right) ; \mathbb{V}_{\tilde{\boldsymbol{u}}^- \sim \tilde{\mathcal{U}}^-} \left( \mathcal{S}(\tilde{\boldsymbol{u}}, \tilde{\boldsymbol{u}}^-) \right) \right\}, \quad (8)$$

where $\sup(\cdot)$ means the upper bound and $\mathbb{V}$ is the variance. We can obviously see that the divergence risk is affected by the mean and variance of the selected negative samples. Yet existing methods [48] usually treat others within the same mini-batch as negative samples for contrastive learning, thereby minimizing the expectation term. Due to the stochasticity of mini-batch sampling, similar samples may be mistakenly chosen as negatives, which increases the variance of negative samples, enlarging the upper bound of the divergence risk.

To tackle it, we propose a weighted complementary contrastive loss as a surrogate objective to achieve divergence risk optimization more robustly, which can be formulated as:

$$\mathcal{L}_{div} = \sum_{i=1}^{b} \sum_{j \neq i}^{b} \underbrace{\exp\left(-\max\left(0, \mathcal{S}(\tilde{\boldsymbol{u}}_i, \tilde{\boldsymbol{u}}_j) - \gamma\right)\right)}_{\text{Weighting term}} \underbrace{\log\left(1 - \mathcal{S}(\tilde{\boldsymbol{u}}_i, \tilde{\boldsymbol{u}}_j)\right)}_{\text{Complementary contrastive term}}, \quad (9)$$

where $\gamma$ is a threshold, above which samples are assigned lower weights. Minimizing the complementary contrastive term optimizes the expectation over negative pairs, while the weighting component can mitigate the impact of high-variance false-negative samples.

Finally, we combine both terms to obtain our loss for domain adaptation tuning, as follows:

$$\mathcal{L} = \lambda \mathcal{L}_{dis} + (1 - \lambda)\mathcal{L}_{div}, \quad (10)$$

where $\lambda$ is a hyperparameter to control the contribution of each component. Minimizing this proxy loss facilitates the reduction of domain adaptation risk, thereby enabling the *retriever* to better adapt to the domain of interaction text.

---

[3]The analysis can be found in our Supplemental Material.

Table 1: Performance comparison on ScanRefer, Nr3D, and Sr3D in terms of R@1, R@5, R@10, and their sum (Rsum). † denotes the use of coarse-grained descriptions as memory.

| Methods | ScanRefer | | | | Nr3D | | | | Sr3D | | | |
|---|---|---|---|---|---|---|---|---|---|---|---|---|
| | R@1 | R@5 | R@10 | Rsum | R@1 | R@5 | R@10 | Rsum | R@1 | R@5 | R@10 | Rsum |
| *w/ coarse-grained descriptions:* | | | | | | | | | | | | |
| VSE∞ (CVPR'21) | 9.7 | 33.1 | 50.2 | 93.0 | 5.8 | 21.5 | 32.5 | 59.8 | 5.5 | 18.9 | 27.4 | 51.8 |
| CHAN (CVPR'23) | 9.4 | 32.3 | 52.1 | 93.8 | 7.5 | 15.0 | 32.1 | 54.6 | 5.4 | 21.3 | 34.8 | 61.5 |
| HREM (CVPR'23) | 10.2 | 34.0 | 51.4 | 95.6 | 7.3 | 18.1 | 31.9 | 57.3 | 5.4 | 21.9 | 33.6 | 60.9 |
| CRCL (NeurIPS'23) | 10.3 | 32.4 | 49.8 | 92.5 | 8.1 | 22.5 | 33.2 | 63.8 | 4.9 | 19.5 | 31.9 | 56.3 |
| RoMa (TMM'25) | 11.4 | 34.8 | 54.4 | 100.6 | 6.5 | 24.8 | 37.6 | 68.9 | 7.8 | 27.3 | 39.3 | 74.4 |
| IDeal | **16.0** | **42.7** | **59.8** | **118.5** | **11.7** | **34.8** | **50.4** | **96.9** | **10.3** | **30.2** | **48.5** | **89.0** |
| *w/ fine-grained descriptions:* | | | | | | | | | | | | |
| ChatIR† (NeurIPS'23) | 21.8 | 55.4 | 73.1 | 150.3 | 15.6 | 40.3 | 58.1 | 114.0 | 12.1 | 35.9 | 50.3 | 98.3 |
| Rewrite† (ICMR'24) | 17.4 | 47.0 | 63.7 | 128.1 | 17.1 | 31.4 | 45.8 | 94.3 | 12.4 | 28.1 | 40.4 | 80.9 |
| MERLIN† (EMNLP'24) | 31.1 | 68.8 | 83.8 | 183.7 | 21.8 | 55.0 | 71.8 | 148.6 | 14.2 | 42.0 | 60.3 | 116.5 |
| BASELINE: IR† | 29.9 | 68.0 | 83.3 | 181.2 | 18.0 | 51.6 | 60.2 | 129.8 | 14.6 | 39.2 | 55.3 | 109.1 |
| BASELINE SUM† | 34.4 | 69.5 | 85.1 | 189.0 | 22.5 | 55.2 | 67.5 | 145.2 | 16.4 | 41.5 | 62.1 | 120.0 |
| IDeal† | **37.8** | **71.8** | **86.4** | **196.0** | **26.4** | **62.7** | **78.7** | **167.8** | **20.2** | **43.1** | **63.1** | **126.4** |

# 4 Experiments

## 4.1 Experimental Setting

**Datasets, baselines, and evaluation metrics**: We adopt the ScanNet 3D scene set along with several description sets (*i.e.*, ScanRefer [51], Nr3D [52], Sr3D [52], and SceneDepict-3D2T [6]) to conduct experiments, where ScanRefer, Nr3D, and Sr3D are employed as query sets, and SceneDepict-3D2T is employed to simulate fine-grained memory. To verify the superiority of our IDeal, we introduce eleven comparative baseline methods: five *conventional offline cross-modal matching methods* (*i.e.*, VSE∞ [45], CHAN [53], HREM [54], CRCL [37], and RoMa [6]), three *interactive cross-modal retrieval methods* (*i.e.*, ChatIR [8], Rewrite [41], and MERLIN [16]), and two *additional strong interactive baselines* (IR and SUM). More specifically, the Iterative Reranking (IR) involves multi-round interaction, where the results are iteratively re-ranked based on the response of each round. The Summary reranking (SUM) also involves interaction, but ultimately aggregates all answers into a comprehensive description for matching.

In addition, we follow [55, 56] to report R@1, R@5, R@10, and their summation (Rsum) as the evaluation metrics. Due to the space limitation, more details of datasets, prompts, and additional experiments are provided in the Supplemental Material.

**Implementation details**: All methods are implemented in PyTorch and carried out on GeForce RTX 3090 GPUs. We adhere to the experimental settings of [6] for all method implementations. We adopt widely-used DGCNN [57] and BERT [58] to obtain fine-grained features for 3D point clouds and texts, respectively. To implement interaction, we explore two approaches to constructing memory:

Table 2: Performance comparison on ScanRefer and Nr3D in terms of R@1, R@5, R@10, and their sum. +*IDeal* indicates plugging the model into our IDeal. † denotes the use of fine-grained descriptions as memory.

| Methods | ScanRefer | | | | Nr3D | | | |
|---|---|---|---|---|---|---|---|---|
| | R@1 | R@5 | R@10 | Rsum | R@1 | R@5 | R@10 | Rsum |
| VSE∞ | 9.7 | 33.1 | 50.2 | 93.0 | 5.8 | 21.5 | 32.5 | 59.8 |
| +IDeal | 13.3 | 38.9 | 57.6 | 109.8 | 8.7 | 27.5 | 42.1 | 78.3 |
| VSE∞† | 14.9 | 42.3 | 61.5 | 118.7 | 16.4 | 47.5 | 55.2 | 119.1 |
| +IDeal† | 35.8 | 70.6 | 85.0 | 191.4 | 21.2 | 52.1 | 68.4 | 141.7 |
| CRCL | 10.3 | 32.4 | 49.8 | 92.5 | 8.1 | 22.5 | 33.2 | 63.8 |
| +IDeal | 13.4 | 35.5 | 56.1 | 105.0 | 7.4 | 25.4 | 38.3 | 71.1 |
| CRCL† | 17.5 | 45.1 | 58.3 | 120.9 | 13.4 | 44.5 | 51.5 | 109.4 |
| +IDeal† | 31.7 | 66.9 | 83.5 | 182.1 | 15.8 | 50.4 | 64.4 | 130.6 |
| RoMa | 9.7 | 33.1 | 50.2 | 93.0 | 8.3 | 27.9 | 37.2 | 73.4 |
| +IDeal | 16.0 | 42.7 | 59.8 | 118.5 | 11.7 | 34.8 | 50.4 | 96.9 |
| RoMa† | 16.7 | 44.8 | 61.6 | 123.1 | 17.4 | 48.5 | 57.5 | 123.4 |
| +IDeal† | 37.8 | 71.8 | 86.4 | 196.0 | 25.4 | 60.7 | 75.7 | 161.8 |

**1)** *Coarse-grained description*: We leverage an LLM to generate rich expansions of queries, serving as memory without introducing any additional information leakage. **2)** *Fine-grained description*: In line with existing interactive methods [43, 13], we simulate the user's memory in real-world scenarios using fine-grained scene descriptions, albeit with access to partial additional information. In our experiments, we utilize Qwen-7B-Instruct [59] as our investigated LLM for the interaction experiments.

## 4.2 Comparison on Text-3D Scene Retrieval

Table 1 presents a comparison between our IDeal and conventional and interactive cross-modal matching methods under two memory settings. Table 2 further demonstrates the performance gains brought by integrating our interactive framework into conventional single-round methods. These results could yield the following observations: **1)** Even without additional fine-grained information, our IDeal achieves competitive performance, highlighting its ability to uncover complementary information implicitly embedded in the queries, gradually alleviating inherent query limitations. **2)** Compared to existing interactive methods, our IDeal also achieves superior performance with access to fine-grained descriptions. This demonstrates that our interactive questioning and retrieval strategies enable an ongoing understanding of user requirements and support a comprehensive interpretation of complex scenes. **3)** Our IDeal can be seamlessly integrated into conventional cross-modal retrieval methods and yields substantial performance gains under both memory settings. This suggests that, beyond equipping offline models with interactive capabilities, IDeal empowers them to more effectively comprehend complex descriptions through interaction.

## 4.3 Ablation Study

In this section, we conduct an ablation study to evaluate the contribution of each proposed component to our IDeal. Specifically, we first ablate the router in the *questioner*, restricting it to ask either $\mathcal{Q}_1$ or $\mathcal{Q}_2$ continuously. In addition, we remove each of the three retrieval prediction strategies, and we examine removing CoT prompting in the reconstruction in the proposed *retriever*. Finally, we investigate the effect of not using the IAT strategy for domain adaptation and sequentially ablate its two loss terms. The results in Table 3 lead to the following observation: **1)** Removing or replacing any component from IDeal results in performance degradation, highlighting the contribution of each component. Specifically, the adaptive questions generated by our *questioner* facilitate a meticulous

Table 3: Ablation studies for components of our IDeal on ScanRefer. RSum is the sum of R@1, R@5, R@10. *w/o* stands for without use.

| Configurations | | ScanRefer | | | |
|---|---|---|---|---|---|
| | | R@1 | R@5 | R@10 | Rsum |
| Questioner | *w/o* $\mathcal{Q}_1$ | 36.0 | 71.2 | 86.7 | 193.9 |
| | *w/o* $\mathcal{Q}_2$ | 26.3 | 61.5 | 77.3 | 165.1 |
| Retriever | *w/o* $\hat{\boldsymbol{p}}_1(\boldsymbol{u}_i)$ | 35.2 | 70.1 | 86.5 | 191.8 |
| | *w/o* $\hat{\boldsymbol{p}}_2(\bar{\boldsymbol{u}}_i)$ | 28.1 | 63.3 | 80.6 | 172.0 |
| | *w/o* $\hat{\boldsymbol{p}}_3(\boldsymbol{s}_i)$ | 31.8 | 67.8 | 84.2 | 183.8 |
| | *w/o* CoT | 35.7 | 71.5 | 85.9 | 193.1 |
| Adaptation | *w/o* IAT | 16.6 | 48.4 | 64.4 | 129.4 |
| | *w/o* $\mathcal{L}_{dis}$ | 34.9 | 69.5 | 84.4 | 188.8 |
| | *w/o* $\mathcal{L}_{div}$ | 35.1 | 69.4 | 84.1 | 191.7 |
| Full | IDeal | 37.8 | 71.8 | 86.4 | 196.0 |

and comprehensive understanding of scenes. The various feature aggregation strategies in the *retriever* contribute to precise scene matching. **2)** Removing or substituting IAT components consistently leads to performance degradation, underscoring the necessity of text domain alignment and adaptation risk minimization in our IAT.

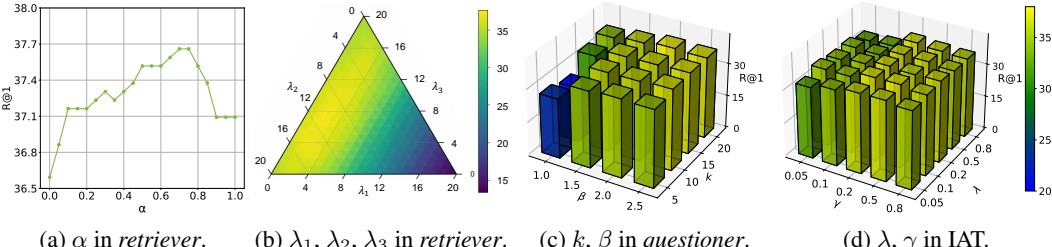

(a) $\alpha$ in *retriever*.    (b) $\lambda_1, \lambda_2, \lambda_3$ in *retriever*.    (c) $k, \beta$ in *questioner*.    (d) $\lambda, \gamma$ in IAT.

Figure 3: PTM performance in terms of R@1 versus different values of the parameters of our IDeal on ScanRefer. (a) and (b) display $\alpha$, $\lambda_1$, $\lambda_2$, and $\lambda_3$ in our *retriever*. (c) shows $k$ and $\beta$ in our *questioner*. (d) shows $\lambda$ and $\gamma$ in IAT.

## 4.4 Parameter Analysis

To evaluate the sensitivity of our IDeal to different hyperparameter settings, we plot the retrieval performance versus different values on ScanRefer, as shown in Figure 3. The experimental results lead to the following observation: **1)** For our *retriever*, tuning greater weights to interactive and reconstruction predictions helps achieve a well-balanced trade-off that fully leverages the interactive responses. Additionally, a higher feature fusion weight (*e.g.*, $\alpha = 0.75$) represents a emphasis on

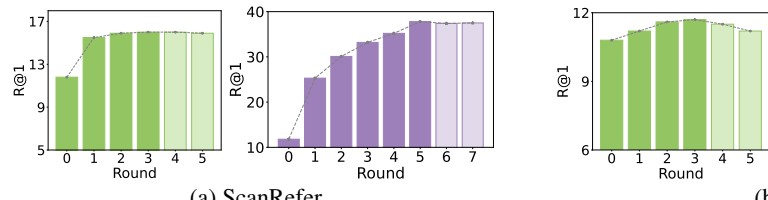

(a) ScanRefer        (b) Nr3D

Figure 4: Performance (R@1) versus rounds on two datasets. Round 0 indicates the setting without interaction. The green and purple bars represent the cases with coarse-grained and fine-grained memory descriptions, respectively. The *lighter* bars indicate no performance gain.

the integration of discriminative features from the refined descriptions, leading to more effective interaction feature fusion. **2)** For our proposed *questioner*, using reasonable and moderate settings of $k$ and $\beta$ (e.g., $k = 20$, $\beta = 2.0$) enables accurate identification of informative descriptions, thereby supporting reasonable decisions on question types in the next round. **3)** During domain adaptation tuning, a relatively wide range of $\lambda$ and $\gamma$ values in IAT (*i.e.*, $\lambda \in [0.2, 0.5]$ and $\gamma \in [0.1, 0.8]$) ensures effective contrastive adaptation and mitigates the impact of false negatives.

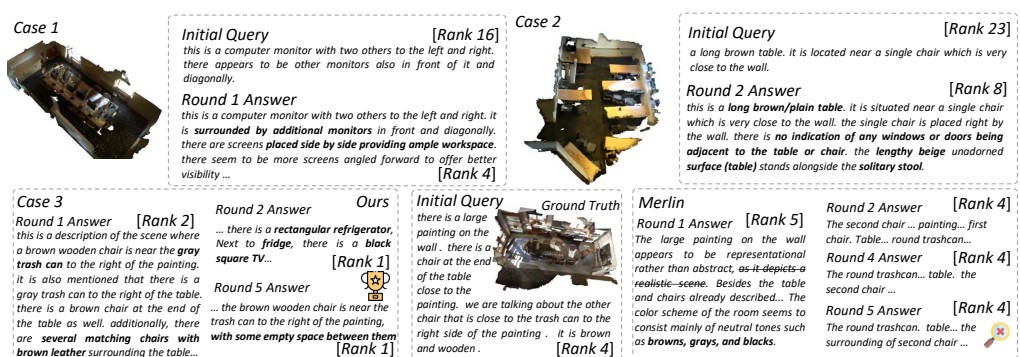

Figure 5: Case illustrations of interactive cross-modal scene matching process of IDeal on ScanRefer. Cases 1–2 and 3 are with coarse-grained and fine-grained memory descriptions, respectively.

## 4.5 Visualization Analysis

To provide a comprehensive analysis of IDeal, we conduct a series of visualization experiments. Specifically, we first present the changes in retrieval performance of IDeal across multiple rounds of interaction, as shown in Figure 4, to analyze the incremental gains brought by each interaction. In addition, we visualize several representative cases, as illustrated in Figure 5. The observations can be drawn from the results: **1)** Interaction consistently improves performance over the first several rounds (five rounds with fine-grained and three rounds with coarse-grained descriptions). Although redundant interactions may inevitably cause performance to saturate or even slightly degrade due to LLM hallucinations or excessively long texts, these results indicate that a reasonable number of interaction steps can effectively enhance the query and improve retrieval performance. **2)** Under the setting with coarse-grained memory texts, IDeal can infer and decompose object attributes and relationships within the queries through interaction, leading to improved retrieval performance. Moreover, with the integration of fine-grained memory, IDeal leverages targeted and switchable questioning to elicit informative responses, continually improving retrieval precision. In contrast, MERLIN [16] frequently generates redundant descriptions confined to local details of scenes.

## 5 Conclusion

In this paper, we propose a novel **I**nteractive Text-3**D** Scene Retrieval Method, namely IDeal, to address the Text-3D Scene Retrieval (T3SR). Our IDeal integrates two components: the Interactive Retrieval Refinement Framework (IRR) and the Interaction Adaptation Tuning strategy (IAT). Specifically, IRR continuously conducts adaptive questioning and comprehensive response fusion,

enabling holistic exploration of 3D scenes for more precise retrieval. IAT performs contrastive domain adaptation for the retriever toward realistic texts, overcoming the performance bottleneck during interaction. Extensive experiments demonstrate the superiority of our IDeal in T3SR task.

**Limitations and Potential Impact Statement:** Although our work has taken the initial step forward in interactive T3SR, there are some limitations and potential impacts that should be acknowledged. First, the performance of the methods is relatively low. Second, we employ LLMs, and more stable and unbiased LLMs and interaction approaches merit further exploration in the future.

# Acknowledgments

This work was supported in part by NSFC under Grant 62472295 and 62372315; in part by the Fundamental Research Funds for the Central Universities under Grant CJ202403; in part by Sichuan Science and Technology Planning Project under Grant 24NSFTD0130, 2024ZDZX0004, 2024NSFTD0049, and in part by the Chengdu Science and Technology Project under Grant 2023-XT00-00004-GX.

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
