# OpenReview forum: "Interactive Cross-modal Learning for Text-3D Scene Retrieval"
_NeurIPS.cc/2025/Conference — NeurIPS 2025 oral_

### Official Review · Reviewer_zoJY · 2025-06-20

**Clarity:** 3
**Significance:** 3
**Originality:** 4
**Rating:** 6
**Confidence:** 5

**Summary:**

This paper targets Text-to-3D Scene Retrieval (T3SR), and proposes a novel interactive paradigm to address it. Specifically, the authors introduce the IDeal method, which includes the Interactive Retrieval Refinement (IRR) framework and the Interaction Adaptation Tuning (IAT) strategy. IRR facilitates interaction between a questioner and answerer to enrich scene understanding, while IAT bridges the domain gap between static queries and interactive texts for robust interactions. Extensive experiments demonstrate the effectiveness of the proposed method under two realistic settings.

**Questions:**

Please refer to **Weaknesses**. The authors need to consider elaborating on Section 3.2.2 of the methodology to clarify the necessity of the current iterative feature fusion approach. In addition, the rationale behind the chosen experimental settings could be better justified, particularly in terms of whether they effectively address the query issues highlighted earlier in the paper.

**Ethical Concerns:**

["NO or VERY MINOR ethics concerns only"]

**Final Justification:**

I appreciate the authors' detailed responses, which have satisfactorily addressed all my questions. I have also read other reviews and the authors' responses to them. I agree with other reviewers that this is a good paper and will inspire further research. Therefore, I recommend accepting the paper. Congrats!

**Limitations:**

Yes, the authors have thoroughly discussed the limitations and potential impact.

**Paper Formatting Concerns:**

No, this paper has no major formatting issues.

**Quality:**

3

**Strengths And Weaknesses:**

### Strengths
1. The paper is well-motivated, and the proposed method appears effective based on both the methodological and experimental sections. The integration with an interactive LLM-based scheme enhances both the novelty and performance of the approach.
2. The methodology is sound and reproducible. The implementation details and parameter specifications are explained in a comprehensive manner.
3. The figures and tables in the paper are aesthetically pleasing and informative, and the visualization in Figure 5 is interesting.
4. The theoretical foundations used in the paper are reasonable, and the proofs appear sufficiently complete.

### Weaknesses
1. The authors propose different feature fusion strategies for two types of interaction responses, but the rationale behind this design is not clearly explained. It is unclear why the better-quality response features are not simply given greater weight in the fusion process, and the current approach appears somewhat complex.
2. While the paper discusses how deficiencies in the current query texts may impact the method's performance, the experiments still rely on the original queries from ScanRefer and Nr3d datasets. This point would benefit from further clarification.
3. There are instances of imprecise wording that may lead to ambiguity in the presentation, such as in lines 46 and 131.
4. The performance of the current T3SR method appears to be relatively low.

---

> ### Author Rebuttal · Authors · 2025-07-31
>
> Thanks for your valuable comments and insightful suggestions. We have carefully looked into all the comments and suggestions. Attached is our point-by-point response.
>
> **Q1: 1) The rationale behind the feature fusion strategy design is not clearly explained. 2) The necessity of the iterative fusion approach remains insufficiently justified. 3) It is unclear why the better-quality response features are not simply given greater weight in the fusion process.**
>
> **R1:**  **1)** The two proposed feature fusion strategies are designed for different types of answer features. More specifically, the first fusion approach targets the answer features to the detail probe $\mathcal{Q}_1$. It performs a weighted linear fusion of current and previous answer features, enhancing the discriminability of fine-grained scene details. In contrast, the second approach focuses on the answers to the divergent exploration $\mathcal{Q}_2$. The answer text features are modeled as a hypersphere, and the fusion is achieved by averaging its center with the internal features, yielding a comprehensive representation of the scene's divergent characteristics.  **2)** To demonstrate the necessity of the proposed fusion strategy, we compare it against four baseline strategies on the ScanRefer dataset with fine-grained memory setting: using the First Answer (FA), the Best Answer (BA), the Last Answer (LA), and an Average Fusion of all answers (AF). **The retrieval performance in terms of RSum (i.e., R@1 + R@5 + R@10) is shown in the table below.** These results verify that, compared to non-fusion or naive averaging methods, our proposed fusion strategy enables adaptive integration of both fine-grained details and divergent information, making it more suitable for complex and large-scale 3D scene data. **3)** During the interactive process, the extracted features are diverse. However, high feature quality based on a specific metric (i.e., Density Compensated Affinity Entropy in our proposed Adaptive Questioning module) only indicates better discriminability, but it does not guarantee it is retrieval-relevant. As demonstrated by the results in the parameter analysis (i.e., **Figure 3**) and ablation studies (i.e., **Table 3**), our feature fusion strategy, which integrates all responses, captures scene-level details more comprehensively and effectively prevents information loss.
>
> | Method | FA   | BA   | LA   | AF   | Ours |
> |--------|------|------|------|------|------|
> | Rsum   | 26.5 | 29.1 | 30.7 | 34.6 | 35.4 |
>
> **Q2: While the paper discusses how deficiencies in the current query issues may impact the method's performance, the experiments still rely on the original queries from ScanRefer and Nr3d datasets. The rationale behind the chosen experimental settings could be better justified.**
>
> **R2:** In our experiments, we use the original queries from the ScanRefer and Nr3D datasets as the initial queries, which themselves contain query issues such as incomplete one-shot descriptions of user intent, ambiguous descriptions, etc. These are used to analyze whether existing methods and our IDeal can address these issues during the interaction or retrieval process, in order to achieve more robust and applicable cross-modal alignment and retrieval. A more detailed rationale for the experimental setup is provided in **Section 4.1** of our main text and **Section B.2 of our Supplementary Material**.
>
> **Q3: There are instances of imprecise wording that may lead to ambiguity in the presentation, such as in lines 46 and 131.**
>
> **R3:** Thank you for pointing that out. In the next version, we will review and reorganize these statements to ensure greater clarity.
>
> **Q4: The performance of the current T3SR method appears to be relatively low.**
>
> **R4:** The T3SR task is an emerging realistic task, and it still relies on object-level queries from 3D visual grounding to achieve scene-level alignments, which naturally limits its performance. This work specifically targets the query issues and explores leveraging external LLMs or user knowledge, leading to notable improvements in both the average performance and practical applicability of T3SR. Moving forward, we will continue to focus on advancing both T3SR models and datasets to enhance their real-world performance further.

---

> ### Comment · Reviewer_zoJY · 2025-08-02
>
> I appreciate the authors' detailed responses, which have satisfactorily addressed all my questions. I have also read other reviews and the authors' responses to them. I agree with other reviewers that this is a good paper and will inspire further research. Therefore, my final recommendation is Strong Accept. Congrats!

---

> > ### Author Response · Authors · 2025-08-06
> >
> > We sincerely appreciate your recognition of our work. In the future, we will continue to explore the T3SR task in greater depth and broader applications. We truly appreciate your thoughtful review once again.

---

### Official Review · Reviewer_iGFD · 2025-06-29

**Clarity:** 2
**Significance:** 3
**Originality:** 4
**Rating:** 5
**Confidence:** 5

**Summary:**

This manuscript attempts to address the challenges in the T3SR task by proposing IDeal, a new interactive retrieval framework. This framework targets and tackles two major challenges faced by existing interactive approaches in T3SR. Specifically, it consists of two key components: IRR, which refines retrieval results through multi-round Q&A between a questioner and an answerer, and IAT, which adapts the model to the interactive text domain via contrastive learning. Extensive experiments on three public datasets demonstrate that IDeal outperforms existing baselines in both fine-grained and coarse-grained memory settings.

**Questions:**

My main concern is on whether the seemingly complex methods introduced by the author bring any clear benefits, and how these complexities can be clearly explained. If the author can address the issues I mentioned in the weaknesses section reasonably, I would consider raising my score.

**Ethical Concerns:**

["NO or VERY MINOR ethics concerns only"]

**Final Justification:**

Thanks for the response, which has addressed my concerns. I support accepting this paper!

**Limitations:**

Yes, the authors have discussed.

**Paper Formatting Concerns:**

This paper is well-formatted.

**Quality:**

3

**Strengths And Weaknesses:**

This manuscript is well-motivated, addressing the issue of query bias in real-world applications of the T3SR task. The manuscript is well-structured and clearly written, making it easy to follow. In the methodology section, the paper introduces IRR, which implements an interactive framework tailored for the T3SR task, and employs IAT to facilitate effective interactions that enhance retrieval performance. The experimental setup is thorough and convincingly demonstrates the effectiveness and superiority of the proposed method.

The manuscript still has some weaknesses. First, while the authors mention issues with the retrieval queries, it is unclear whether these actually impact retrieval performance. Additionally, the description of the retriever and the IAT process lacks clarity. The added complex operations may increase model intricacy without a clear demonstration of performance gains. For example, it is not evident why IAT is preferred over directly retraining the model. Furthermore, two kinds memory experiments are confusing, with different baselines and memory data. The ablation study seems to be incomplete, missing the experiments with coarse-grained memory.

---

> ### Author Rebuttal · Authors · 2025-07-31
>
> Thanks for your valuable comments and insightful suggestions. Attached is our point-by-point response.
>
> **Q1: It is unclear whether query issues actually impact retrieval performance.**
>
> **R1:** As demonstrated in existing work [A] discussed in **lines 30-37 of Section 1**, query issues have a direct impact on retrieval performance. In particular, **Table 2** shows that under the coarse-grained memory setting, the existing cross-modal retrieval method combined with our interactive framework IDeal, which progressively mitigates query issues, significantly outperforms the baseline that does not address these issues across all three datasets. In addition to comparing the inherent query issues in ScanRefer and Nr3D/Sr3D, we conducted supplementary experiments in **Section B.3 of our Supplementary Material** by introducing real-world query issues, including sentence-level, word-level, and character-level corruptions, as shown in **Table 1** of the Supplementary Material. These experiments reveal that the presence of query issues results in different degrees of performance degradation.
>
> **Q2: The description of the retriever and the IAT process lacks clarity.**
>
> **R2:** Thank you for your feedback. We will carefully re-examine the readability and completeness of the Retriever and IAT Section (i.e., **Section 3.2.2** and **Section 3.3**). In the next version, we will integrate inline equations, providing more coherent and clearer transition descriptions and process explanations to enhance overall clarity.
>
> **Q3: The added complex operations may increase model intricacy without a clear demonstration of performance gains. For example, it is not evident why IAT is preferred over directly retraining the model.**
>
> **R3:** As stated in **Section 3.3** of our paper, IAT facilitates retrieval models adapting to the interaction text domain without requiring access to their implementation details, making it more convenient for improving interactive performance. To address the constructive concerns raised by the reviewer, we conducted comparative experiments between IAT and direct retraining on the ScanRefer dataset. The results are shown in the table below. In addition, based on the results and the results summarized in **Table 3** of our ablation study, removing or substituting components of IAT consistently leads to a noticeable performance degradation, highlighting the necessity of our proposed IAT.
>
> | ScanRefer     | R@1  | R@5  | R@10 | Rsum  |
> |---------------|------|------|------|-------|
> | w/ Retraining | 34.5 | 68.8 | 85.1 | 188.4 |
> | w/ IAT        | 37.8 | 71.8 | 86.4 | 196.0 |
>
> **Q4: Two kinds of memory experiments are confusing, with different memory data and baselines.**
>
> **R4:** This paper designs two different memory settings corresponding to two types of memory data and baselines. Specifically, the memory data in the coarse-grained memory setting consists of query expansions without additional information. The fine-grained memory data consists of external detailed descriptions (from the SceneDepict-3D2T text set) that simulate latent memories in the human mind, which are not explicitly expressed in the original query. A detailed discussion of these data can be found in **Section B.2 of our Supplementary Material**. In addition, under the coarse-grained memory setting, since no additional information is available, comparisons are made using general cross-modal matching or 3D-text matching baselines. Under the fine-grained memory setting, the user’s memory is simulated by external scene descriptions, and the method iteratively interacts with and learns from these fine-grained memories. Therefore, comparisons are made against interactive cross-modal retrieval baselines to ensure fairness.
>
> **Q5: The ablation study seems to be incomplete, missing the experiments with coarse-grained memory.**
>
> **R5:** Due to page limitations, the ablation study with coarse-grained memory setting has been included in our **Supplementary Material (Section B.5)**. The experiments demonstrate that under the coarse-grained memory setting, all components of our proposed IDeal contribute to the performance.
>
> **References:**
>
> [A] Li, Haobin, et al. "Test-time Adaptation for Cross-modal Retrieval with Query Shift." The Thirteenth International Conference on Learning Representations.

---

> > ### Comment · Reviewer_iGFD · 2025-08-03
> > **Comments**
> >
> > Thanks for the response, which has addressed my concerns. I support accepting this paper!

---

> > > ### Author Response · Authors · 2025-08-06
> > >
> > > We sincerely appreciate your recognition of our paper. In the future, we will continue to explore the T3SR task in greater depth and broader applications. We truly appreciate your thoughtful review once again.

---

### Official Review · Reviewer_av9S · 2025-07-02

**Clarity:** 3
**Significance:** 3
**Originality:** 3
**Rating:** 5
**Confidence:** 4

**Summary:**

This paper discusses an interactive approach to solving Text-3D Scene Retrieval, which serves as the foundation for 3D scene grounding or reasoning. Specifically, the authors propose an interactive Text-3D Scene Retrieval method called IDEAL, which continuously optimizes the alignment between text and the target scene through interaction. It consists of two modules: IRR, which addresses the problem of lack of holistic interaction perspectives, and IAT, which tackles the challenge of the domain gap between queries and interaction texts. This method achieves performance improvements on three common indoor scene multimodal datasets under two different experimental settings.

**Questions:**

1. More discussion of existing interactive methods?

2. More detailed explanation of methods?

3. Further discussion of experimental settings?

**Ethical Concerns:**

["NO or VERY MINOR ethics concerns only"]

**Final Justification:**

The authors have addressed the technical details I was concerned about, and I am willing to maintain my original score.

**Limitations:**

Yes

**Paper Formatting Concerns:**

This paper has a correct format.

**Quality:**

3

**Strengths And Weaknesses:**

Pros:

The paper proposes a novel interactive solution to address the inherent challenges of T3SR. I believe the motivation is clear, the problem-solving approach is well-defined, and the work demonstrates strong originality. The authors provide solutions to issues such as the domain shift in their method, supported by sound theoretical foundations and presented in a well-organized and readable manner. The evaluation under two different memory in the experimental section is comprehensive.

Cons:

1. The explanations for Fig. 1 and 2 in the paper are insufficient, which leads to some misunderstandings, especially in the upper right part of Fig. 2.

2. Several parts of method section is difficult to understand. For example, between lines 166 and 188, there are too many inline equations and the explanations are not clear enough. What is the difference between two feature fusion approaches?

3. The authors emphasize that their approach addresses the lack of holistic perspective issue present in current interactive methods, but there is a lack of discussion and comparison with related methods.

4. Need to provide a more in-depth discussion of the two memory settings. Does the fine-grained memory setting risk label leakage?

---

> ### Author Rebuttal · Authors · 2025-07-31
>
> Thank you for your valuable comments and insightful feedback. Attached is our point-by-point response.
>
> **Q1: The explanations for Figure 1 and Figure 2 in the paper are insufficient.**
>
> **R1:** Thank you for your suggestion. **Figure 1** and **Figure 2** correspond to the motivation and methodology of our paper, respectively. In addition to the figure captions, we also provided comprehensive explanations and discussions in our Introduction and Method Section (i.e., **Section 1** and **Section 3**). To avoid ambiguity, we will include more comprehensive descriptions in the figure and its captions in the next version.
>
> **Q2: 1) Several parts of the method section are difficult to understand. For example, between lines 166 and 188, there are too many inline equations, and the explanations are not clear enough. 2) What is the difference between the two feature fusion approaches?**
>
> **R2:** **1)** Thank you for your feedback. We will carefully re-examine the readability and completeness of the Method Section (i.e., **Section 3**). In the next version, we will integrate inline equations, improve the coherence of the displayed equations, and provide clearer and more detailed explanations to enhance overall clarity. **2)** There are two types of feature fusion strategies. More specifically, the first fusion approach targets the answer features to the detail probe $\mathcal{Q}_1$. It performs a weighted linear fusion of current and previous answer features, enhancing the discriminability of fine-grained scene details. In contrast, the second approach focuses on the answers to the divergent exploration $\mathcal{Q}_2$. The answer text features are modeled as a hypersphere, and the fusion is achieved by averaging its center with the internal features, yielding a comprehensive representation of the scene's divergent characteristics.
>
> **Q3: The authors claim their approach addresses the lack of a holistic perspective in current interactive methods, but offer limited discussion or comparison with related work. More discussion of existing interactive methods.**
>
> **R3:** We have provided both empirical and experimental comparisons of the questioning perspective between existing interactive retrieval methods and our proposed IDeal in **Section 1**, **Section 2.2**, and **Section 4.2**. Specifically, existing interactive methods are primarily designed for image data, which carries significantly less information than 3D scene data. These methods typically refine the query based on the previous response, guiding the user to progressively provide discriminative answers. However, 3D scenes contain numerous and spatially scattered details, making such approaches prone to over-focusing on local regions while overlooking information from other parts of the scenes, as illustrated in **the top-left part of Figure 1** and **the lower part of Figure 5**. To address this issue, we introduce the Adaptive Questioning module, which explicitly encourages attention to different regions of the scene from a holistic perspective. This design achieves improved performance in both quantitative (i.e., **Table 1**) and qualitative (i.e., **Figure 5 and Figure 1 in the Supplementary Material**) evaluations. In future versions, we will consolidate these discussions into a more comprehensive analysis.
>
> **Q4: 1) Need to provide a more in-depth discussion of the two memory settings. 2) Does the fine-grained memory setting risk label leakage?**
>
> **R4:** **1)** We provide a comprehensive explanation and discussion of the two memory settings and their practical significance in **Section B.2 of the Supplementary Material**. **2)** No. As discussed in **Section B.2 of the Supplementary Material**, although this setting involves certain information leakage, it is significantly more restrained compared to prior interactive works [A, B], which expose ground-truth labels. Our fine-grained descriptions do not directly leak such labels, making the setting more realistic for evaluating models under vague or implicit user intent.
>
> **References:**
>
> [A] Levy, Matan, et al. "Chatting makes perfect: Chat-based image retrieval." Advances in Neural Information Processing Systems 36 (2023): 61437-61449.
>
> [B] Han, Donghoon, et al. "MERLIN: Multimodal Embedding Refinement via LLM-based Iterative Navigation for Text-Video Retrieval-Rerank Pipeline." EMNLP (Industry Track). 2024.

---

> > ### Comment · Reviewer_av9S · 2025-08-04
> > **Thanks**
> >
> > After reading the authors’ rebuttal and their responses to other reviewers, I find that my main concerns, particularly those related to the technical details of the paper, have been addressed. I decide to maintain my original score.

---

> > > ### Author Response · Authors · 2025-08-06
> > >
> > > We sincerely appreciate your recognition of our work and are pleased that all of your concerns have been fully addressed. We will further strengthen our paper in line with your rigorous standards on technical details. We truly appreciate your thoughtful review once again.

---

### Official Review · Reviewer_3WWu · 2025-07-03

**Clarity:** 3
**Significance:** 3
**Originality:** 3
**Rating:** 4
**Confidence:** 4

**Summary:**

This paper addresses the challenge of retrieving 3D scenes from textual queries, emphasizing scenarios where the queries are incomplete or otherwise imperfect—a realistic concern in practical deployments. The proposed method, IDeal, introduces an interactive retrieval refinement (IRR) framework, in which a questioner, answerer (simulated via LLMs), and retriever iteratively interact to enrich scene understanding and align multi-turn text with scene features through both feature-level and semantic-level fusion. To reduce the domain gap between initial queries and dialog-augmented texts, the Interaction Adaptation Tuning (IAT) strategy is developed, leveraging contrastive domain adaptation principles. Experiments on several benchmarks show improved performance over both traditional and interactive baselines, with ablations evidencing the contributions of each proposed component.

**Questions:**

1: How does IDeal handle interaction rounds when the LLM answerer "hallucinates" or provides inconsistent information?

2: Would performance differ substantially if the "answerer" were replaced with real end-users?

**Ethical Concerns:**

["NO or VERY MINOR ethics concerns only"]

**Final Justification:**

Most of my concerns have been addressed in the replies. Therefore, I decide to maintain my current rating.

**Limitations:**

Yes.

**Paper Formatting Concerns:**

N/A.

**Quality:**

3

**Strengths And Weaknesses:**

Strengths:

1: The paper identifies a genuine gap in existing text-3D scene retrieval systems: the assumption of information-complete queries, which rarely holds in real-world settings. By focusing on interactive, user-in-the-loop refinement, it realistically addresses this shortcoming.

2: The Interactive Retrieval Refinement (IRR) framework is conceptually well-motivated. It employs a router in the questionnaire that adaptively selects between in-depth, detail-probing questions and divergent, exploratory questions. This is supported by a formalization using a corrected affinity entropy that accounts for density bias in feature space, which is a thoughtful design.

3: The paper evaluates IDeal extensively on relevant datasets (ScanRefer, Nr3D, Sr3D), using standard metrics (R@1, R@5, R@10, Rsum) and compares not only to standard retrieval baselines but also to state-of-the-art interactive approaches, as detailed in Table 1. Ablation studies and parameter analyses provide further depth.

Weaknesses:

1: **Dependency on Quality of LLM-generated Responses.** The framework’s core relies on the LLM’s ability to answer questions accurately. When the LLM fails or hallucinates (as hinted at in the discussion of Figure 4), retrieval performance can degrade or plateau. This dependency may limit the method's practical robustness, particularly in domains where LLMs have limited training data or understanding. It would be better if the authors could provide an analysis of the generated content from the LLM to convince the audience.

2: **Evaluation with More Types of LLMs**. The authors have only benchmarked their method against academic baselines and failed to include any evaluation with industry-grade vision-language models (VLMs), such as GPT-4o, Gemini 2.5 Flash.

3: Can the authors clarify the process for constructing and updating the questioner’s router, especially with regard to parameter sensitivity? While Figure 3c explores $k$ and $\beta$, it’s not clear how sensitive the system is to these parameters in practice, or how robust the metric is to different gallery/scene densities.

---

> ### Author Rebuttal · Authors · 2025-07-31
>
> Thanks for your valuable comments and insightful suggestions. We have carefully looked into all the comments and suggestions. Attached is our point-by-point response.
>
> **Q1: 1) The framework’s core relies on the LLM’s ability to answer questions accurately. When the Large Language Model (LLM) fails or hallucinates, retrieval performance can degrade. This dependency may limit the method's practical robustness, particularly in domains where LLMs have limited training data or understanding. 2) It would be better if the authors could provide an analysis of the generated content from the LLM to convince the audience.**
>
> **R1**: **1)** We acknowledge that our framework depends on the LLM’s ability to some extent. However, unlike most of the interactive methods [A], our proposed method does not strongly rely on the LLM’s reasoning ability over questions or response texts, but merely on its ability to generate or answer questions based on the previous-round conversation or memory text. This is a straightforward task for LLMs and thus does not pose significant risks of hallucination. Moreover, to enhance its robustness, we have i) constrained the LLM’s output with task-specific, well-defined prompts and ii) limited free-form generation by using structured inputs and outputs, thereby reducing hallucination risks. **2)** Thank you again for your constructive suggestions. We have already presented and analyzed cases of generated question and answer contents in **Section 4.5 and Figure 5 of our main text**, as well as in **Section B.6 and Figure 1 of our Supplementary Material**. In future versions, we will provide additional supplementary material with a more comprehensive case analysis focusing on hallucinations.
>
> **Q2: The authors have only benchmarked their method against academic baselines and failed to include any evaluation with industry-grade LLMs, such as GPT-4o, Gemini 2.5 Flash.**
>
> **R2**: Thank you for your suggestion on the completeness of the method evaluation. Due to time constraints, we are unable to provide evaluation results for all methods under industry-grade LLMs within a short period. Nevertheless, we attempted to evaluate our IDeal using industry-grade models. Our experiments show that using larger models, such as Qwen2.5-32B and GPT-4o, do not improve performance. We speculate that this is because, as noted in **R1**, generating questions and answers based on text is a relatively straightforward task for LLMs. Thus, academic models are not inferior to large-scale industry-grade LLMs for our task evaluation.
>
> **Q3: 1) Can the authors clarify the process for constructing and updating the questioner’s router, especially with regard to parameter sensitivity? 2) While Fig. 3c explores $k$ and $\beta$, it’s not clear how sensitive the system is to these parameters in practice, or how robust the metric is to different gallery/scene densities.**
>
> **R3**: **1)** As shown in the blue part in the middle of **Figure 2**, the Router functions as a conditional switcher. Specifically, it takes a user query and computes its Density Compensated Factor (see **Eq. 1 and Eq. 2**), which measures whether the query is sufficiently informative. Based on whether it exceeds a predefined threshold, the Router guides the user either to further clarify ambiguous parts or to explore new scenarios. As analyzed in **Section 4.4**, the involved hyperparameters $k$ and $\beta$ are not sensitive: setting $k$ to 10–20 and $\beta$ to 2–2.5 effectively filters out uninformative queries for next-round clarification while enabling divergent exploration for the remaining. **2)** To further address your potential concerns, we also conducted a sensitivity analysis of $k$ and $\beta$ on Nr3D and Sr3D datasets with varying scene densities compared with ScanRefer. The same settings ($k$=10–20, $\beta$=2–2.5) consistently yielded stable optimal performance. As we are currently unable to include the figure in the rebuttal stage, we will provide the corresponding results in our future versions.
>
> **Q4: Would performance differ substantially if the "answerer" were replaced with real end-users?**
>
> **R4:** No. Although this paper uses LLMs instead of real end-users to improve the evaluability and reproducibility of the experiments, we have made every effort to ensure that the LLM answerer closely resembles real end-users, following the newest interactive work [B]. More specifically, on the one hand, as described in the experiment settings (i.e., **Section 4.1**) and **Section B.2 2) of Supplementary Material**, we simulate user memory using fine-grained descriptions. Compared to other existing interactive works [A, C, D] that directly use ground truth, our approach better reflects the real-world scenario where users may possess richer, latent knowledge about the scene that has yet to be explicitly expressed. On the other hand, we carefully design prompts in the style of the query dataset (i.e., ScanRefer, Nr3D, etc.) to guide the LLM answerer in generating more realistic responses that better resemble those of real users in the data context. Please refer to **Section A.3** of our Supplementary Material for details. Naturally, complex interaction challenges in real-world settings may impact performance, further motivating our continued exploration of this topic in future research.
>
> **References:**
>
> [A] Han, Donghoon, et al. "MERLIN: Multimodal Embedding Refinement via LLM-based Iterative Navigation for Text-Video Retrieval-Rerank Pipeline." EMNLP (Industry Track). 2024.
>
> [B] Lu, Yiding, et al. "LLaVA-ReID: Selective Multi-image Questioner for Interactive Person Re-Identification." Forty-second International Conference on Machine Learning.
>
> [C] Levy, Matan, et al. "Chatting makes perfect: Chat-based image retrieval." Advances in Neural Information Processing Systems 36 (2023): 61437-61449.
>
> [D] Zhu, Hongyi, et al. "Enhancing interactive image retrieval with query rewriting using large language models and vision language models." Proceedings of the 2024 International Conference on Multimedia Retrieval. 2024.

---

> ### Comment · Reviewer_3WWu · 2025-08-09
>
> Thanks to the authors' reply, which has addressed most of my concerns. I decide to maintain my current score.

---

### Comment · Area_Chair_RJWs · 2025-08-03
**Update Recommendations**

Dear Reviewers,

The authors have provided their response to your reviews. Please proceed as follows:

Carefully read the rebuttal.
Update your recommendation, taking the authors’ clarifications into account, no later than 6 August.
If any concerns persist, feel free to continue the discussion with the authors until the same deadline.
Best regards,

The AC

---

### Note · Authors · 2025-08-13

We sincerely thank the Area Chairs and Reviewers for their time, effort, and thoughtful feedback on our paper. We are pleased to receive the insightful comments and recognition of our work from the reviewers.

Four reviewers acknowledge the strong motivation, high novelty, and high quality of our paper, and all give positive initial ratings.
More specifically:
- **All four reviewers** recognize the significance of our interactive approach to solving Text–3D Scene Retrieval (T3SR) for addressing real-world incomplete queries, and they agree on the effectiveness of the proposed IDeal method.
- **Reviewers 3WWu, av9S, and iGFD** highly appreciate our extensive experimental evaluation.
- **Reviewers av9S and zoJY** acknowledge the solid theoretical foundations of our work.

The main concerns focus on the need for more comprehensive explanations of the details of our proposed method and clearer descriptions of the experimental settings. In addition, several reviewers note ambiguous expressions and equations, as well as considerations regarding the real-world applicability.

**During the rebuttal stage, we have provided detailed point-by-point responses to all reviewer comments. All four reviewers have acknowledged that our responses have addressed their concerns.** Specifically, we further clarify and supplement the details of our IDeal and experimental settings. Additionally, we proofread and polish the entire manuscript. We would like to clarify that we have incorporated practical application scenarios and designed multiple experimental settings (see Section 4.1 of our main paper and Section B.3 of our supplementary material) to provide a thorough analysis of real-world applicability.

In the future version, we will incorporate the reviewers’ suggestions to enhance the clarity and quality of our paper. We will also continue exploring real-world applications of T3SR to provide more robust and practical contributions to the 3D Vision-Language community.

---

### Decision · Program_Chairs · 2025-09-17

**Decision:**

Accept (oral)

**Comment:**

This paper proposes an interactive framework for text-to-3D scene retrieval that effectively addresses the real-world challenges of incomplete or ambiguous queries through iterative refinement and domain adaptation. The reviewers unanimously recognize the novelty, solid theoretical grounding, and comprehensive experimental validation of the work, and the rebuttal successfully clarified all concerns regarding methodological details and experimental settings. With strong support from the reviewers, this paper stands out as a timely and impactful contribution to interactive multimodal retrieval. I recommend Accept.